# Communication, inclusion and psychological wellbeing among deaf and hard of hearing children: A qualitative study in the Gaza Strip

Nathaniel Scherer[1]*, Tracey Smythe[1,2], Ramadan Hussein[3], Lorraine Wapling[4], Shaffa Hameed[1], Julian Eaton[5,6], Naim Kabaja[3], Ritsuko Kakuma[5], Sarah Polack[1]

1 International Centre for Evidence in Disability, London School of Hygiene & Tropical Medicine, London, United Kingdom, 2 Division of physiotherapy, Department of Health and Rehabilitation Sciences, Stellenbosch University, Stellenbosch, South Africa, 3 Atfaluna Society for Deaf Children, Gaza City, Palestinian Territories, 4 International Disability Research Centre, University College London, London, United Kingdom, 5 Centre for Global Mental Health, London School of Hygiene & Tropical Medicine, London, United Kingdom, 6 CBM Global Disability Inclusion, London, United Kingdom

* nathaniel.scherer@lshtm.ac.uk

**Data Availability Statement:** Excerpts from the transcripts of this qualitative research have been made available within the article. Providing

## Abstract

Deaf and hard of hearing children are at risk of exclusion from community life and education, which may increase their risk of mental health conditions. This study explores the experience of deaf and hard of hearing children in the Gaza Strip, with particular focus on the factors that contribute to psychological wellbeing and distress. In-depth interviews were conducted with 17 deaf and hard of hearing children, 10 caregivers of deaf and hard of hearing children and eight teachers of deaf and hard of hearing children in mainstream and special schools, across the Gaza Strip. Further, three focus group discussions were held with deaf and hard of hearing adults and disability leaders, mental health specialists and other teachers of deaf and hard of hearing children. Data collection was completed in August 2020. Key themes identified in the analysis included lack of accessible communication, community exclusion, negative attitudes towards hearing impairment and deafness and the impact on deaf and hard of hearing children's sense of self, and limited family knowledge on hearing impairment and deafness. Further findings focused on strategies to improve the inclusion of deaf and hard of hearing children and how to promote wellbeing. In conclusion, participants in this study believed that deaf and hard of hearing children in the Gaza Strip are at increased risk of mental health conditions. Changes are needed across community and government structures, including education systems, to promote the inclusion of deaf and hard of hearing children and to support their psychological wellbeing. Recommendations from the findings include increasing efforts to improve awareness and reduce stigma, providing better access to sign language for deaf and hard of hearing children, and offering training for teachers of deaf and hard of hearing children, especially in mainstream environments.

anonymised excerpts (i.e. quotations) in publicly-available publications was confirmed by participants during the informed consent process. However, full-transcripts are not available via a public data repository. Sharing full-transcripts, even with known identifiers removed, was not approved by the ethics committees and was not confirmation sought from participants during the informed consent procedure. For queries, please contact the Corresponding Author, or the LSHTM Research Ethics Committee via email (ethics@lshtm.ac.uk).

**Funding:** This research was funded by the German Federal Ministry for Economic Cooperation and Development (BMZ; Bundesministerium für wirtschaftliche Zusammenarbeit und Entwicklung) through CBM International under the project "Improving the psychosocial resilience of children with and without disabilities in the Gaza Strip" to NS, SP, JE, RH and NK. The funders had no role in study design, data collection and analysis, decision to publish, or preparation of the manuscript. Authors NS and RH received salary through this funding for the research undertaken.

**Competing interests:** The authors have declared that no competing interests exist.

# Introduction

One in five people experience hearing impairment, of whom more than 80% live in low- and middle-income countries (LMICs) [1]. Whilst hearing impairment is more common in older persons, estimates indicate that there are 70 million children with hearing impairment globally, of whom 34 million have moderate to profound hearing impairment [1].

Deaf or hard of hearing children across the world can experience challenges in daily life, with regards to educational attainment, social inclusion and community participation [2–4]. Deaf and hard of hearing children are more likely than hearing peers to report poorer quality of life in school and social domains [3]. These negative life experiences are associated with a negative impact on emotional wellbeing and mental health, and there is increased risk of mental health conditions among deaf and hard of hearing adult and child populations, including anxiety and depression [5–13].

## Context and aim of the study

Little research exists on the experiences of deaf and hard of hearing children in LMICs. The vast majority of research comes from high-income settings and findings are not representative of LMICs, where there exist different conceptions and attitudes towards disability and hearing impairment, and often more limited availability of specialised support and services. Further, little evidence is available on the experiences of deaf and hard of hearing in conflict-affected settings, where, in general, people are at greater risk of mental health conditions [14]. Deaf and hard of hearing children in conflict-affected settings may face double-jeopardy to their psychological wellbeing, both from the stressors experienced by many deaf and hard of hearing children and those that arise from being affected by conflict, such as trauma and lack of access to services [15, 16].

This study explores the experiences of deaf and hard of hearing children in the Gaza Strip, an area that has been affected by conflict and blockade, resulting in a deteriorating socioeconomic situation that has negatively impacted on the economy, education, food security, and access to basic services [17]. There is some evidence to suggest that the Gaza Strip and the West Bank has a high prevalence of hearing impairment among infants and children [18, 19]. Historically, this has resulted from high instances of consanguinity, although this practice is becoming less common [18]. However, there is scarce research available on the lived experience of deaf and hard of hearing children in the Gaza Strip, including information on experiences in school and the wider community.

Ongoing conflict, economic hardship and restricted life prospects are associated with high rates of depression, anxiety and post-traumatic stress disorder for youth in the Gaza Strip [20–25]. These stressors make it difficult for children and adolescents to develop a strong sense of self and to develop aspirations for the future [25]. Despite mental health services being available in the Gaza Strip, many young people face organisational and cultural barriers to access, and services often do not meet their need [20, 25]. Nevertheless, research has suggested that education and family support have improved wellbeing of youth in the Gaza Strip, providing inspiration and building self-esteem [25]. Although there is evidence on mental health among youth generally, there is limited information on factors that contribute to psychological wellbeing among deaf and hard of hearing children in the Gaza Strip.

Given the dearth of evidence on deaf and hard of hearing children in the Gaza Strip, this study aimed to investigate the experiences of deaf and hard of hearing children in daily life and the factors that contribute to their inclusion, quality of life and mental health. This information can help contribute to the development of context appropriate policies and interventions.

## Materials and methods

This study used qualitative research methods to explore the experiences of deaf and hard of hearing children in the Gaza Strip. In-depth interviews were conducted with 35 participants, including 17 primary school age deaf and hard of hearing children, 10 caregivers, and eight teachers of deaf and hard of hearing children in mainstream and special schools. Three focus group discussions were held with five deaf and hard of hearing adults and disability leaders, seven mental health and psychosocial support specialists and school counsellors, and 13 teachers of deaf and hard of hearing children in mainstream and special education settings.

The research and analysis were guided by a steering committee of key stakeholders in the Gaza Strip. This group comprised: five deaf and hard of hearing people and representatives from organisations of deaf and hard of hearing people; two caregivers of deaf and hard of hearing children; four teachers of deaf and hard of hearing children; three mental health and psychosocial support specialists; and five representatives from government level institutions, including the Ministry of Education.

### Study context

The research was conducted across each region of the Gaza Strip. In the Gaza Strip, deaf children with severe to profound hearing impairment are typically taught in special schools. Hard of hearing children with mild to moderate hearing impairment are typically taught in mainstream schools. Mainstream schools are government run and free to attend. Special schools are not supported by government financing and are often managed by non-governmental organisations (NGOs). Families pay a fee for their child to attend a special school. Each mainstream and special school is required to have a school counsellor who is tasked with promoting mental wellbeing and supporting children experiencing distress. When a child presents with severe symptoms of psychological distress, they are referred by the school to counsellors at district level who provide more in-depth assessment and support. At this district level, there are mental health specialists trained to work with deaf and hard of hearing children.

### Participants

A summary of participant characteristics is presented in S1 Appendix.

Participants for the in-depth interviews were purposively sampled through mainstream and special schools across four regions in the Gaza Strip (North, Gaza City, Middle, South) in order to maximise variation, as based on Patton's maximum variation sampling strategy [26]. Participants were selected for variation across age, sex, severity of hearing impairment and type of school attended (mainstream or special school). To aid the sampling strategy, we segmented the target group by the criteria listed above, aiming for equal representation in each category. From this, we developed a minimum sample size of 32, with additional participants included if topics required further investigation [26]. In total, 35 participants were included in the in-depth interviews, comprised of 17 deaf and hard of hearing children aged 6–12, 10 caregivers of deaf and hard of hearing children, and eight teachers of deaf and hard of hearing children in mainstream and special schools. Severity of hearing impairment was confirmed by an audiologist. Hearing profiles, assistive technology usage and communication mode of the deaf and hard of hearing children are available in S1 Appendix.

Participants for the three focus group discussions were recruited through local organisations of persons with disabilities (OPDs), disability NGOs, mental health associations and schools. The first group comprised of three deaf and hard of hearing adults and sign language users, one individual with a physical impairment and one individual with a visual impairment, who also works as a counsellor. These individuals were representatives from OPDs and NGOs

supporting deaf and hard of hearing people. The second group comprised five school counsellors and two mental health professionals working with children in the community. The third included seven special school teachers and six mainstream school teachers. These teachers were not the same as those included in the in-depth interviews.

## Data collection

Data collection was conducted in July 2020. Interviews were held in a trusted, central location, as requested by participants, many of whom expressed concern about being audio-recorded in their own home. As the research was conducted during the COVID-19 pandemic, government guidance on safety was followed.

Semi structured, in-depth interviews with deaf and hard of hearing children, caregivers and teachers sought to explore the views, experiences, emotions and perspectives of deaf and hard of hearing children and caregivers in the Gaza Strip, both in school and the wider community. In addition, focus group discussions were held with individuals who have expertise in understanding the experiences and needs of deaf and hard of hearing children and their families. Through dynamic group discussion, these focus groups aimed to elicit multiple opinions on the experiences of deaf and hard of hearing children, as well as recommendations to improve quality of life and psychological wellbeing. Topic guides are available in S2 Appendix.

The interview guides were developed by the research team, piloted with six participants (two children, two caregivers, two teachers) and adapted as needed thereafter. Questions explored the experiences of deaf and hard of hearing children, their families and their teachers, focusing on experiences in education, healthcare and community life. Specific questions were asked on communication, mental health, wellbeing and attitudes towards hearing impairment and deafness. Topic guides for the focus group discussions aimed to further understand these areas, explore the wider context for deaf people in the community, and appraise the factors contributing to wellbeing. Data was collected during the COVID-19 pandemic, however the study sought to understand experiences in more typical times, and we did not explicitly explore the experiences of deaf and hard of hearing children and families in the context of the pandemic.

Interviews were conducted in Arabic with those using oral communication. Palestinian sign language interpretation was available for those using sign language for communication. Emotion cards were used to help children understand a question or to describe their feelings. Children were interviewed individually, unless the child or caregiver desired otherwise. There was occasionally need for a family member to sit in as interpreter, when a child communicated through a signed language developed with family at home. Each interview lasted between 30–60 minutes. Five pilot interviews were conducted, in which to practice interviewing techniques and refine the topic guides. The data from these interviews has been included in the analysis. All interviews were audio-recorded. Sign language users had their response reported orally by an interpreter. Audio-recordings were transcribed verbatim. Each was translated and transcribed directly into English, by an independent transcriber, and subsequently checked against the audio file by the interviewer. Each transcript was anonymised and stored on a secure server.

## Data analysis

Data was analysed using thematic analysis [27]. In step one, NS familiarised themself with the data, noting down initial ideas for coding. In step two, NS developed a coding framework, which was discussed and refined with RH, TS and SP. NS subsequently coded transcripts in NVivo 12 using this coding framework, which was piloted and iteratively adapted throughout

analysis. In step three, emerging themes were identified and refined by NS, comparing relationships between codes and across groups. In step four and five, NS reviewed these themes with RH, RK, TS and SP, mapping the themes against the data set and the emerging narrative. Where RH, RK, TS and SP had comments and queries, NS reviewed the coding and themes, refine and recategorizing codes, where needed. In the final step, participant quotes and narratives were extracted for the report. The themes, narrative and recommendations were discussed with the steering committee, before producing the final report.

### Reflexivity

Interviews were conducted by a trained interviewer (RH), who received a three-day, one-to-one training with the lead author (NS), a qualitative researcher in global disability and mental health. The interviewer's background was as an experienced audiologist at a major service unit for deaf and hard of hearing people in the Gaza Strip; Atfaluna Society for Deaf Children. The interviewer knew some local sign language, but not enough to conduct interviews without support from an interpreter. The interviews themselves were held at Atfaluna's headquarters, at the request of participants. With Atfaluna known to many deaf and hard of hearing people and their families in the Gaza Strip, their responses may have been influenced by the dynamic. To mitigate this, we assured participants that their responses and data would be kept confidential and that their access to services would not be affected as a result of the interview. For many, Atfaluna offers a safe space for deaf and hard of hearing children and families, and in many instances, the interviewer reported the benefit of this association on building comfort and rapport.

The analysis was led by NS, who is based in the United Kingdom. NS supported the data collection remotely through co-training, regular post-interview discussions with the interviewer and remote participation in workshops/advisory committees as travel to Gaza strip was not possible due to the COVID-19 pandemic. To limit potential biases in analysis led by an off-site foreign researcher, the interpretations were reviewed by the research team in the Gaza Strip with experience in supporting deaf and hard of hearing children in the region. The findings were further discussed with the steering committee in the Gaza Strip.

### Reliability and validity

To promote reliability, validity and rigour of this qualitative research, we adopted a number of methodological strategies, as recommended in the literature [28, 29]: (1) Reflexivity, as we've described above, our efforts to reflect on our position as researchers sought to provide transparency on our biases and the influence of their on the analysis; (2) Validation of findings was achieved by presenting our initial analysis to the steering committee and groups of deaf and hard of hearing adults, to ensure that our interpretation of the findings were appropriate to the context in the Gaza Strip; (3) Peer debriefing was utilised throughout coding and analysis between NS and co-authors, to ensure exploration into all relevant themes; (4) Triangulation was core to the methodology, with authors adopting multiple data sources and data collection methods, to strive for a comprehensive exploration of this topic; (5) Transparency of procedures was achieved by keeping a paper trail of methodology, records and explanations, and presenting these in the manuscript.

### Ethical considerations

Ethical approval for this study was obtained from the Research Ethics Committee at the London school of Hygiene & Tropical Medicine (19144) and the Palestinian Health Research Council (PHRC/HC/697/20).

Informed written consent was received from all participants. Caregivers or guardians provided informed consent for children and further assent was sought from children using a simplified information sheet.

## Results

Across the interviews, there was the consistent understanding from adult participants that deaf and hard of hearing children in the Gaza Strip were at increased risk of psychological distress and mental health concerns. Various associated factors were reported, including language deprivation, community attitudes and family support.

## Lack of accessible communication and language deprivation

Participants reported that deaf and hard of hearing children with milder levels of hearing impairment were less likely to experience challenges with communication and community participation, and they were often able to communicate orally. However, deaf and hard of hearing children with moderate to profound hearing impairment faced daily challenges with communication and engagement with others in the community as sign language use is limited across the region. Communication challenges restricted social inclusion and participation, negatively impacting on the psychological wellbeing of deaf and hard of hearing children.

Although Palestinian sign language use is increasing across the Gaza Strip, it is still limited in healthcare, mainstream schools and community settings. Teachers, caregivers and deaf and hard of hearing adults told us how many deaf and hard of hearing children experience language deprivation, with families unable to teach them sign language or refusing to allow their child to learn.

"I: Do they use sign language?

P: No it is not allowed. . . once her uncle tried to use sign language and I prevented him from doing so. If I wanted her to learn and use sign language, I would give her sign language courses."

(Father of a child aged 10–12 with severe hearing impairment)

We were told that many families in the Gaza Strip do not know Palestinian sign language and do not use it at home, even if their child uses the language at school and despite many of the children interviewed telling us that they prefer to communicate using sign language. Many of the families interviewed perceived sign language negatively and desired their child to use oral speech, even when the child had difficulty doing so. Some attributed sign language for their child's poor oral skills and some did not understand it to be a language itself and their child's natural form of communication. Limited sign language use and support resulted in examples of family and child having limited communication or regular misunderstanding and miscommunication.

"If their cousins gather and talk, she will withdraw because she cannot hear and she cannot understand what they are saying, as they are talking fast. The same thing when my sisters and I gather. This thing bothers me and bothers her, but what we can do?"

(Mother of a child aged 10–12 with moderate hearing impairment)

The lack of engagement with sign language was not the case for all families, and there were positive examples of caregivers supporting the use of sign language by their child and seeking

training for themselves. After learning themselves, there were examples told to us of family members training others, including hearing siblings, aunts, uncles and cousins. Support for sign language and accessible communication was often seen in families with multiple deaf and hard of hearing children, where siblings could communicate with and support one another. One child interviewed even taught their hearing friends at a mainstream school some simple signs, so they could better communicate and bond. However, despite wishing to support sign language use, some family members reported limited awareness on where to receive sign language training and communication support. Some children used home sign language as a result of limited exposure to Palestinian sign language and can only communicate with a few others as a result; often deaf and hard of hearing siblings from whom they learned to communicate.

## Communication and social inclusion

Some children interviewed had strong oral skills. They found it easier to participate in community life and were integrated into life at a mainstream school. That said, many of these children still expressed a preference for sign language, because oral communication was often tiring and difficult to maintain, especially in group situations and noisy environments.

Many deaf and hard of hearing children in mainstream schools described difficulty in keeping up with teachers and peers, who often have limited awareness of deaf and hard of hearing children's communication needs and how best to support them. We were told by many, including teachers, that difficulty with communication in mainstream schools significantly hampered the learning and education for many deaf and hard of hearing children. Limited communication also makes inclusion difficult with hearing peers in school and the community. This was reported to cause isolation and psychological distress. Some caregivers told us that they do not allow their deaf or hard of hearing child out into the community because of their difficulties communicating with others.

> "His nature is sociable, he tries to communicate with other kids and join them, but they cannot understand what he is saying, which makes him upset."

> (Father of a child aged 6–9 with moderate hearing impairment)

In contrast, the special school environment was more supportive of sign language and children and teachers in these settings reported very few instances of miscommunication or isolation of deaf children. A deaf adult told us how, as a child, he had to move back to a special school after joining a mainstream school, as communication in the mainstream environment was too challenging and teachers didn't account for his deafness and communication needs. For example, he lipreads and teachers regularly faced away from him when talking. He expressed regret that he had to leave, as he wanted to learn in a mainstream environment.

Language deprivation, challenges in communication and resulting social isolation were noted by many of the participants as the major influence on a child's mental health and wellbeing.

> I: Are deaf children at risk of mental health issues?

> P: Yes, because they cannot keep up with the family talking. What's causing a psychological problem is that a family will shorten up a long conversation into two words. This upsets a child and makes them prefer loneliness.

> (Deaf adult and representative from an organisation of persons with disabilities, communicating via a sign language interpreter)

Participants, including deaf adults, teachers and mental health specialists, discussed communication in the context of protecting and promoting mental health and wellbeing. Communication and the resulting social integration were described as the main source of a child's confidence, self-esteem and wellbeing. Participants recommended that families and teachers be given sign language training to combat the language deprivation experienced by many deaf and hard of hearing children in the Gaza Strip and to promote their psychological wellbeing.

"When the student has good communication in sign language with their family members, this comforts and relieves them very much. When the mother communicates in sign language with her child, the child becomes able to express and explain everything. The same thing happens when they communicate with their teacher, they feel that they are just as other classmates. When they go on a school journey and they communicate with teachers and other people, they break the ice and overcome the fear with outside. As a result, they become more confident. . . This gradually strengthens their personalities and their self-confidence."

(Teacher in a special school)

Deaf adults told that us that communication was improving in the Gaza Strip, with Palestinian sign language more widely used than when they were children. However, improvements in the attitude towards and understanding of sign language was slow, and many families were still resistant to its use. Services and programmes to support use of sign-language, as well as speech and language therapy and occupational therapy, were often unavailable to caregivers, except through one or two specialist organisations. Foremost was Atfaluna Society for Deaf Children, which was often cited as the primary support for deaf children in the Gaza Strip, although not all participants were aware of the available support from this organisation and other specialist organisations supporting deaf and hard of hearing children. They did not have information on support available, where to seek support and how to access it. Deaf adults also told us that it was difficult to find qualified sign language interpreters in the region and sign language training was sparse. The services were not sufficient to meet the need.

"Frankly, the Department of Education hasn't given us sign language courses. I use simple, easy things for students. For example, go, come, and understand. . . I had four students with hearing disabilities and I couldn't communicate well with them."

(Teacher in a mainstream school)

## Assistive technology and communication

Assistive technology was commonly discussed in the context of communication. Many caregivers described hearing aids as useful for children who use them and many wanted their child to use one, even if the improvements in hearing were marginal. However, some caregivers and children spoke of ill-functioning or ill-fitting hearing aids, negating any benefits for communication and limiting their ability to engage in community and classroom activities. Caregivers told us that paying for and maintaining the hearing aids is expensive, with some caregivers forgoing food to afford batteries. Repair services are also limited and there can be long waitlists if a hearing aid is damaged, especially if originally sourced from a country out of the Gaza Strip. Cochlear implants were perceived positively among the participants, with some having gone abroad to receive them, to countries such as Egypt and Australia. However, this was not available for all families of deaf and hard of hearing children. Many of those that had done so still

relied on financial support from an NGO or international sponsor. Cochlear implantation is becoming more widely available in the Gaza Strip and many families note this as an appealing option to them. Deaf adults interviewed, however, were often disparaging of such assistive technologies and encouraged families to embrace their child's hearing impairment and to cement a strong Deaf identity in them.

## Community attitudes impacting confidence and identity

Stigma and discrimination in the community also presented barriers to social inclusion and participation for deaf and hard of hearing children, and had a negative impact on psychological wellbeing. This was not the case for all deaf and hard of hearing children and adults, and some reported positive relationships with neighbours and hearing peers. However, we were told that many in the community still held discriminatory views and behaviours. Awareness of disability and deafness was said to be poor. Knowledge was often based on a medical or charity model of disability and deafness, with little understanding of rights-based perspectives in line with the United Nations Convention on the Rights of Persons with Disabilities (UNCRPD) [30]. This limited awareness was seen most commonly among mainstream school teachers, who often had very little knowledge of deafness and supporting deaf and hard of hearing children in their classroom.

Respondents felt that it was common for the community to look on families and deaf and hard of hearing children with sadness and pity, and we were told that deaf people were seen to be unequal or were simply not acknowledged in society.

"For example, if we are in a car, people will stare at him [her deaf son], and some of them ask, 'Why is he like this?'. Someone said, 'Poor boy'. I said, 'Why? What's wrong with him [her deaf son]? Thank God, they are a gift'."

(Mother of child aged 6–9 with severe hearing impairment)

Participants reported that negative attitudes often proceed to more overt acts of discrimination. We were told that hearing people often use derogatory terms when talking about deaf people, including an Arabic slur which translates loosely to 'dumb' or 'retarded' (derogatory terms themselves across much of the world).

"They whisper and point at me that I'm dumb [Arabic slur]"

(Child aged 10–12 with severe hearing impairment)

Discrimination was said to severely impact on deaf and hard of hearing children (and adults) in the Gaza Strip. Children and caregivers told us of the psychological distress caused when community members or other children made them feel different or marginalised. Children told us they often cried as a result. There were several examples of such instances. A father told us how he moved his family to another region of the Gaza Strip, as his son was facing such severe discrimination; others used to steal his hearing aids and run away. Similarly, a mother told us how her daughter was bullied constantly at school.

"She was bullied at school by other children. They would remove her hearing aid and bully her. She would come home crying. I tried to speak to the teachers about this, but they didn't help. No one understands her communication or possibly needs. I decided to move her to another school but the same thing happened. I eventually moved her to a special school,

where things got better. She has since joined a mainstream government school after her communication improved."

(Mother of child aged 10–12 with moderate hearing impairment)

There were a few examples reported in which bullying and discrimination dissipated once caregivers and teachers explained deafness to other children, other teachers and community members. Some caregivers reported speaking about deafness to teachers individually, as well as presenting to other caregivers and children at the school. Some teachers with experience and knowledge about deafness instructed other teachers at their school, with support from the school counsellor and headteacher. These efforts were reported to improve awareness of the experience of deaf and hard of hearing children and helped dissipate some of the stigma and discrimination seen. Caregivers are important advocates in these initiatives.

"It was hard at the beginning. They mocked him every single day, which made him sad. He was crying all the time. I explained his case to the children and now they understand him and they started playing with him."

(Mother of child aged 6–9 in a mainstream school with moderate hearing impairment)

Special schools offered a more inclusive and supportive environment, with no discrimination reported to us amongst teachers or other children. However, there were no reports of systematic governmental or organisational training and resources for special school teachers or others to instruct mainstream teachers on supporting the inclusion of deaf and hard of hearing children in mainstream schools.

Further, few deaf and hard of hearing spaces and community structures were reported. Participants advocated for more deaf and hard of hearing friendly spaces, where deaf and hard of hearing children could learn about their hearing impairment, Deaf culture, sign language and be engaged with other deaf and hard of hearing people in a safe space. Participants also suggested that deaf and hard of hearing children need to be more included in community activities, to reduce isolation and improve psychological wellbeing.

"For us as deaf people, not more than 3% are included in society. At the present time, the community should make more effort to give deaf people their own space, and to involve them in community activities, so that people know more about the skills and experiences of deaf people."

(Deaf adult and representative from an organisation of persons with disabilities, communicating via a sign language interpreter)

## Family knowledge and internalised stigma

It was clear that many caregivers and families interviewed were proud of their deaf and hard of hearing children and expressed to us deep love, but some had not fully accepted their child's hearing impairment and deafness. Caregivers were said to occasionally hide their deaf or hard of hearing child, so as not to affect their siblings' future and experience in the community. Others did not want to see their child wearing a hearing aid and refused to let them learn sign language. A deaf adult told us of a story in which a mother refused for her deaf son to marry a deaf woman, as she feared their child would also be deaf; she wanted him to marry a hearing woman instead.

Limited acceptance seemed to result from limited understanding and awareness of deafness and the rights of deaf and hard of hearing people, paired with the influence of negative community attitudes. Many of the caregivers interviewed had never received information or support on hearing impairment when their child was diagnosed, despite wishing for such. Caregivers called for structured training programmes with other caregivers, access to online resources and support from specialist organisations. Those that did receive such support reported positive benefits and typically presented with more positive attitudes to hearing impairment.

For some other caregivers, internalised stigma among families appeared to result from the challenges they perceive a result of having a deaf or hard of hearing child.

"It affects all my life system. It takes all of me, my efforts, my time, and it takes me from my other children."

(Mother of child aged 6–9 with severe hearing impairment)

As a result of negative community and family attitudes, some children told us that they did not want to be deaf or hard of hearing, or to be different, and many were embarrassed and ashamed to use sign language in public or to be seen wearing a hearing aid. One father said his daughter feared she would never get married because of her hearing aid.

"Once she said, 'I know why they are treating me like that, it is because I wear a hearing aid.' And one time I saw her crying because her sister got engaged and she said, 'I will never get married because of my hearing aid'."

(Father of child aged 10–12 with severe hearing impairment)

The discrimination in the community, and the internalised stigma among families and children, negatively impacts on deaf and hard of hearing children's sense of identity and ultimately their mental health and wellbeing. Children were said to get upset and angry at being different to others. One mother told us of how she enrolled her daughter in psychosocial counselling, as she was very down and tired of being "different".

## Building a sense of self and wellbeing

Participants believed community awareness and improving understanding of deafness and hearing impairment was of the utmost importance in addressing stigma, discrimination and improving engagement with Deaf culture and norms, such as sign language. Addressing stigma and discrimination was said to be vital in improving the psychological wellbeing of deaf and hard of hearing children. Some told us that this awareness is continually improving in the Gaza Strip, particularly among the younger generation, who are more knowledgeable and supportive of deaf and hard of hearing people, as a result of greater exposure to the internet and international campaigns.

Children that engaged with deaf and hard of hearing peers and Deaf communities reported the positive impact on quality of life and psychological wellbeing. They enjoyed having deaf and hard of hearing friends and interacting with others with a shared experience. Many deaf and hard of hearing children in mainstream education said that they had few deaf and hard of hearing friends, although they would like to. Those that did have deaf and hard of hearing friends valued those relationships. "The first is to focus on things that help to integrate them into society, and the second is to spread culture about hearing disability among

children so that they accept deaf colleagues or any other disability without making fun of them."

(Adult with a mobility impairment and representative from an organisation of persons with disabilities)

Building a sense of self and building self-esteem was said to be key in promoting mental health and wellbeing among deaf and hard of hearing children. We were told that building a sense of identity and confidence in children starts with the family. Participants told us how important it was for families to learn about hearing impairment, deafness, deaf and hard of hearing rights, communication, sign language and how best to support their child. Many participants believed that deaf and hard of hearing children develop a strong sense of self and mental wellbeing when families are accepting and supportive of their children. Participants noted that caregivers must promote belief and expectation in their child.

Further, we were told of the importance of caregiver involvement in their child's education and life experience at school, especially in mainstream education. Doing so was considered to improve awareness and inclusion at schools, and caregivers expressed eagerness to work closely with teachers to ensure consistent communication and support.

Examples of good practice were seen in some special schools, including awareness raising and psychosocial support. This included sessions with caregivers on how to communicate with and support their deaf or hard of hearing child. Further, we were recommended that deaf and hard of hearing role models visit schools, both to raise awareness amongst the students on deafness and inclusion and to help build the self-esteem and aspirations of deaf and hard of hearing children and their families.

"We have a lot of attitudinal problems with adults and teenagers. We keep implementing awareness raising activities and psychosocial support sessions. These are very important to be included, especially as caregivers lack sign language, so they don't communicate with their children. Teachers manage the whole communication process and guide the students. We also implement awareness raising activities with caregivers to guide them on how to communicate with and treat their children. We advise them to talk to their children, to understand them, and to identify and analyse their problems."

(Teacher at a special school)

## Promoting deaf and hard of hearing inclusion

As presented, participants reported a number of factors that impact on deaf and hard of hearing children's mental health and psychological wellbeing. Intervention and support is needed across a number of areas to promote wellbeing, including reducing stigma and improving communication. Deaf adults and teachers emphasised that the development of support for deaf and hard of hearing children and families at school and in the community is not going to happen without government support. According to the deaf adults we spoke with, inclusion at government level was viewed as talk with no action. We were told that this stemmed from limited understanding of hearing impairment and deafness at government level, which, in turn, contributed to the lack of inclusion and awareness in the community.

"Even the government decision makers know nothing about us [deaf people]. What we are, what we feel and what we need. They don't know us. There's no awareness of what disability

really is. They think we deaf people are less intelligent than others, they think our learning abilities are slow and hopeless... We, like all other disabilities, are no different and no less."

(Deaf adult and representative from an organisation of persons with disabilities, communicating via a sign language interpreter)

Deaf and hard of hearing children in the Gaza Strip were reported to be facing increased risk of mental health issues. Efforts are needed to build self-esteem and identity in deaf children, so that they may reach their potential and realise personal wellbeing.

"When a child with disability is left behind because of their disability, this would complicate their mental health and cause them lots of psychosocial problems. We need to improve their skills, look into their talents and work on developing them. This would improve their self-trust and help them overcome their mental health and psychosocial problems."

(School counsellor at a mainstream school)

For this to occur, families, teachers and the community need the understanding on how to support inclusion of deaf and hard of hearing children, which includes knowledge on communication and deaf culture.

## Discussion

Participants in this study reported that deaf and hard of hearing children in the Gaza Strip are at increased risk of psychological distress, resulting from language deprivation, discrimination and exclusion from community participation. These findings are consistent with previous literature on the social inclusion and mental health of deaf and hard of hearing populations [5–13].

Children in special school environments commonly used sign language and engaged with deaf and hard of hearing peers. Often, their caregivers were well-informed on the support available, on hearing impairment, on deafness and on deaf and hard of hearing rights, as a result of support and outreach from the special schools. These children appeared to be at lower risk of poor mental health. Children in mainstream environments were most likely to feel included if they used oral communication. These children talked of having hearing friends and generally positive experiences in the community. They tended to have milder hearing impairment and/or used a hearing aid or cochlear implant. They did not often report experiences of mental distress, likely related to fewer challenges in daily life and feeling part of the wider community. As is consistent with previous literature, our interviews indicated that the children most at risk of poor mental health were those experiencing language deprivation and social exclusion, especially in a mainstream schooling environment [31–33]. Often these deaf and hard of hearing children had moderate or severe hearing impairment, yet families did not support them to learn sign language, despite their limited capacity for oral communication. Many faced challenges with learning in school, including making friends. These children appeared to live with a sense of self that is in conflict; neither part of the hearing community and neither part of the Deaf community. This is consistent with previous literature on social identity and mental health among deaf and hard of hearing populations, whereby deaf and hard of hearing people associating with neither hearing or d/Deaf groups can lack a strong sense of self, negatively impacting their mental health [31, 32, 34, 35]. In contrast, mental health and wellbeing is often better in deaf and hard of hearing people using oral language and who associate with the hearing community, or those who have a strong sense of Deaf identity and use sign language. Deaf identities and wellbeing is a nuanced area and one with limited

literature. We recommend additional research in the future into these topics, especially in LMICs and conflict affected populations.

Our findings suggest that, in general, Deaf culture and awareness is limited in the Gaza Strip. Many deaf and hard of hearing children do not have a positive self-image and many deaf and hard of hearing children in the Gaza Strip experience prejudice and discrimination at some point in the community. This was not universal, with some describing positive experiences. Largely, prejudice and stigma appeared to stem from a lack of awareness on hearing impairment and deafness, and understanding of Palestinian sign language as a language. This lack of awareness and knowledge was viewed as a challenge at both community and government level, and was shown in school settings. Stigma is a common barrier experienced by youth with disabilities, but the current state of evidence on stigma-reduction activities for children with disabilities in LMICs is limited [36]. Most studies are of low methodological quality and as a result, current stigma reduction interventions are not evidence-based and often strategies do not target different levels of stigma [36]. That said, there are some promising stigma-reduction interventions for children with disabilities identified in recent systematic reviews, including community education and social contact [36]. Recommendations from additional evidence reviews include the active involvement of people with disabilities (including youth) and caregivers in the development and delivery of stigma-reduction interventions and interventions that empower people with disabilities [37, 38]. Evidence on interventions is rarely focused solely on deaf and hard of hearing children, and there must be appropriate development efforts in partnership with deaf and hard of hearing children and families if looking to implement a deaf and hard of hearing stigma-reduction intervention in the Gaza Strip.

Key in our findings is the language deprivation experienced by some deaf and hard of hearing children in the Gaza Strip. Language deprivation in early life is a fundamental issue and has lifelong consequences [39]. Beyond isolation and loneliness, language deprivation will negatively impact on a child's key developmental skills [39]. Language is also the foundation of an individual's culture and social identity. Depriving deaf and hard of hearing children the right to natural sign language limits the progression of deaf and hard of hearing children's social identity, their self-esteem and their mental wellbeing. In the Gaza Strip, there are a number of factors contributing to this language deprivation. First, is the lack of opportunities for early exposure to sign language for deaf and hard of hearing children, training services for families and a lack of sign language interpreters. The infrastructure is not currently able to meet the demand. Second, this study indicated is the lack of family commitment to sign language. This was not the case for all families in this study, but it was evident in many. The reluctance to allow deaf and hard of hearing children to learn sign language appeared to be related to community stigma and limited information and awareness both on deafness and the fact the sign language is a language. In addition, limited training infrastructure resulted in limited opportunities to learn sign language, limited motivation to do so and limited knowledge on the value of sign language. Addressing inaccessible language and communication (and language deprivation in early life) should be a priority for supporting deaf and hard of hearing people in the Gaza Strip. Promoting Palestinian sign language use across the region will promote the inclusion of deaf and hard of hearing children and will help develop stronger deaf and hard of hearing communities. In their position paper and charter on the language rights of deaf children, the World Federation of the Deaf outlined actions that governments across the world must take, in order to promote the right of deaf children to fully develop their cultural and linguistic identity, as enshrined in the United Nations Convention on the Rights of Persons with Disabilities [40, 41]. They call for governments to "implement programmes to support the teaching of sign language to family members and carers of deaf children, in cooperation with Deaf communities and deaf sign language teachers." The position paper provides specific evidence and

action-based recommendations for governments and community institutions to follow, including the need for early exposure to sign language and promoting family support. As part of the programme in which this research has been placed, CBM (an international disability NGO) and Atfaluna Society for Deaf Children have developed opportunities to learn Palestinian sign language for over 1,000 deaf and hard of hearing children and their families in the Gaza Strip.

As well as improved access to Palestinian sign language, many deaf and hard of hearing children would benefit from improved access to oral-communication services and devices. Our findings reveal a need to improve access to assistive technology, including hearing aid and cochlear implant assessment, provision and maintenance. Many respondents in this study reported experiencing difficulties with poorly functioning assistive products, which hindered oral communication for those that could use and may prefer it. In order to provide early support and intervention, the government in the Gaza Strip may wish to provide systematic newborn hearing screening programmes. Regardless of a family's decision on the use of hearing aids, cochlear implants and audiology therapies, we encourage health professionals to immediately refer deaf and hard of hearing children to Deaf associations, networks of parents of deaf and hard of hearing children and schools that cater for the needs of deaf and hard of hearing children when they first identify hearing loss.

In addition to the above recommendations, training for families may be important to improve deaf and hard of hearing awareness and mitigate the risk of language deprivation. Evidence shows that early intervention with caregivers promotes better awareness on disability, rights and supporting the child [42]. Interventions may include structured caregiver training programmes, such as the Juntos programme, which comprises interventions based on 10 module programme, delivered by caregivers with lived experience to a group of new caregivers and caregivers [43–45]. These programmes help caregivers learn the skills to support their child, whilst also offering peer support and sense of community, although these programmes have not been evaluated with families of deaf and hard of hearing children. Other similar interventions include the parent-mediated World Health Organization Caregiver Skills Training programme [46]. The use of social media may also be important to connect caregivers, share information on deafness and develop community.

With the importance of education and the role that school plays in the inclusion and wellbeing of deaf and hard of hearing children, it is important that schools are be able to support deaf and hard of hearing children. Throughout the interviews, it was clear that special schools in the Gaza Strip were better equipped to provide this support to deaf and hard of hearing children, with Palestinian sign language being routinely used to communicate, aiding both education and social inclusion. With the promotion of inclusive education across the world, mainstream schools in the Gaza Strip need the resources and training to better support deaf and hard of hearing children in their classrooms, including sign language training and interpretation services [47]. The relative merits of segregated and inclusive school systems are widely debated [48]. The findings of this study highlight the importance of strengthening inclusion in mainstream schools, so that deaf and hard of hearing children can attend if they wish. Although special schools in the Gaza Strip may continue to be needed, and may be preferred by some children and families, deaf and hard of hearing children should have the option to attend a mainstream school should they wish, with facilities available to support their full integration, in line with the principles of inclusive education. There is need for structures and resources by which mainstream school administrations and teachers can receive support and training to deliver inclusive education for deaf and hard of hearing children, whether this be from special school teachers or other certified personnel. Training for teachers should also include information on mental health and psychosocial support for deaf and hard of hearing

children in their classrooms, including information on promoting wellbeing, recognising symptoms of distress, and actions to take when they identify an issue. As presented in this research, central to promoting the wellbeing and inclusion of deaf and hard of hearing children at school is partnership with caregivers, teachers and the child themselves. This promotes targeted solutions to support a child, both at school and home, and it gives the platform for caregivers to help teachers improve deaf awareness and inclusion in their classrooms. Until mainstream schools in the Gaza Strip are equipped to support all deaf and hard of hearing children, there is still need for a twin-track approach, with special schools still available and well-resourced.

In efforts to support deaf and hard of hearing children, there is need for government, funders and service providers to partner with deaf and hard of hearing communities and organisations of deaf and hard of hearing people in the Gaza Strip. Evidence has shown that organisations of persons with disabilities can produce positive outcomes for people with disabilities and these organisations are key in achieving equal rights for deaf and hard of hearing people and people with disabilities [49]. In the Gaza Strip, this may include training government, communities and families on deaf and hard of hearing rights and inclusion, building confidence for participation among deaf and hard of hearing children and families, and developing networks of deaf and hard of hearing communities. These organisations are often restricted by limited financial and human resources and they may therefore benefit from capacity building and increased funding to scale-up their work and effectively partner with government and other stakeholders looking to support deaf and hard of hearing children [49].

There is also a need to develop mental health support that is targeted to the needs of deaf and hard of hearing children. Although important to address the risk factors of poor mental health, as identified in this study, it is equally important to ensure appropriate support for those that do experience distress. In a systematic review of mental health support for deaf and hard of hearing children, appropriate interventions included peer-support, caregiver-child interaction therapy and resilience training [50]. Many interventions can be provided in schools and adaptations necessary include appropriate communication, including sign language. Mental health service provision for deaf and hard of hearing people would benefit from recruiting signing mental health professionals, deaf and hard of hearing professionals, and sign language interpreters specialized in mental health. Throughout, it is imperative to consult and coordinate with deaf and hard of hearing children and families, to ensure feasible, acceptable and sustainable interventions.

Children and adolescents in the Gaza Strip are at increased risk of mental health conditions, regardless of disability [25]. deaf and hard of hearing children face additional stressors, as demonstrated in this study. They can be said to experience double-jeopardy. They face stressors to their mental health and wellbeing as a result of living in a challenging region of the world and they face stressors as a result of living in an environment that is not fully inclusive and supportive of deaf and hard of hearing people. These stressors will intersect and raise unique challenges for deaf and hard of hearing children. This population is thus a priority for support.

Interesting to note in this discussion is the limited information reported by participants on the direct impact of conflict and blockade on the psychological wellbeing of deaf and hard of hearing children. Few participants reported on these issues, instead focusing on the topics discussed in the results of this study. This may be because questions asked in the topic guides did not include conflict-specific questions. Questions asked more broadly on factors that influence the mental health of children in the Gaza Strip. There was opportunity for participants to mention conflict and blockade under these questions. It is important to note also that data collection occurred before the recent escalation of rocket fire in 2021 and there is need to assess the

wellbeing of deaf and hard of hearing children (and indeed all children) since this time, so that appropriate support can be provided.

## Limitations

This research encountered challenges resulting from the COVID-19 pandemic. These affected the process of data collection and the involvement of the lead researcher in the Gaza Strip itself. Several checks were in place to strengthen the integrity of data and interpretations, including remote training and supervision, and discussion of the findings with a steering committee, including people with lived experience.

It is also worth noting the greater number of female caregivers interviewed, compared to male. Fathers and other male caregivers were more difficult to recruit, as most were busy during working hours. Moreover, female caregivers in the Gaza Strip are most typically the primary caregiver for deaf and hard of hearing children and when we attempted to recruit to male caregivers, many told us that the mother or female caregiver would be able to give better information than they.

## Conclusion

Deaf and hard of hearing children in the Gaza Strip experience a number of stressors that are detrimental to their mental health and wellbeing, including language deprivation, inaccessible communication and discrimination. Families, civil society actors and education providers must be empowered with the knowledge, skills and resources to provide an inclusive and supportive environment to deaf and hard of hearing children.

## Supporting information

**S1 Appendix. Participant characteristics.**
(DOCX)

**S2 Appendix. In-depth interview and focus group discussion topic guides.**
(DOCX)

**S3 Appendix. Questionnaire on inclusivity in global research.**
(DOCX)

## Acknowledgments

Thank you to all of the participants who generously gave their time to take part in the study.

## Author Contributions

**Conceptualization:** Nathaniel Scherer, Tracey Smythe, Julian Eaton, Naim Kabaja, Ritsuko Kakuma, Sarah Polack.

**Data curation:** Nathaniel Scherer.

**Formal analysis:** Nathaniel Scherer, Tracey Smythe, Ritsuko Kakuma, Sarah Polack.

**Funding acquisition:** Nathaniel Scherer, Julian Eaton, Naim Kabaja, Sarah Polack.

**Investigation:** Nathaniel Scherer, Tracey Smythe, Ramadan Hussein, Ritsuko Kakuma, Sarah Polack.

**Methodology:** Nathaniel Scherer, Tracey Smythe, Ramadan Hussein, Lorraine Wapling, Shaffa Hameed, Ritsuko Kakuma, Sarah Polack.

**Project administration:** Nathaniel Scherer, Ramadan Hussein.

**Resources:** Nathaniel Scherer, Ramadan Hussein.

**Software:** Nathaniel Scherer.

**Supervision:** Tracey Smythe, Julian Eaton, Naim Kabaja, Ritsuko Kakuma, Sarah Polack.

**Visualization:** Nathaniel Scherer.

**Writing – original draft:** Nathaniel Scherer.

**Writing – review & editing:** Tracey Smythe, Ramadan Hussein, Lorraine Wapling, Shaffa Hameed, Julian Eaton, Naim Kabaja, Ritsuko Kakuma, Sarah Polack.

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
