## [Decision Letter · Decision Letter 0]

6 Mar 2023

PGPH-D-22-01952

Communication, inclusion and psychological wellbeing among deaf and hard of hearing children: A qualitative study in the Gaza Strip

Dear Dr. Scherer,

Thank you for submitting your manuscript to PLOS Global Public Health. After careful consideration, we feel that it has merit but does not fully meet PLOS Global Public Health’s publication criteria as it currently stands. Therefore, we invite you to submit a revised version of the manuscript that addresses the points raised during the review process.

Your manuscript has been assessed by two expert reviewers, whose comments are appended below. The reviewers are generally positive about your study, but have made several requests for additional information on the methodology and discussion which will improve the reproducibility and clarify the implications of your study. Please ensure you respond to each point carefully in your response to reviewers document, and modify your manuscript accordingly.

We look forward to receiving your revised manuscript.

Kind regards,

Dr Joseph Donlan

Staff Editor

Journal Requirements:

2. Please include a complete copy of PLOS’ questionnaire on inclusivity in global research in your revised manuscript. Our policy for research in this area aims to improve transparency in the reporting of research performed outside of researchers’ own country or community. The policy applies to researchers who have travelled to a different country to conduct research, research with Indigenous populations or their lands, and research on cultural artefacts. The questionnaire can also be requested at the journal’s discretion for any other submissions, even if these conditions are not met.  Please find more information on the policy and a link to download a blank copy of the questionnaire here: https://journals.plos.org/globalpublichealth/s/best-practices-in-research-reporting. Please upload a completed version of your questionnaire as Supporting Information when you resubmit your manuscript.

Additional Editor Comments (if provided):

Reviewers' comments:

Reviewer's Responses to Questions

**Comments to the Author**

1. Does this manuscript meet PLOS Global Public Health’s publication criteria? Is the manuscript technically sound, and do the data support the conclusions? The manuscript must describe methodologically and ethically rigorous research with conclusions that are appropriately drawn based on the data presented.

Reviewer #1: Yes

Reviewer #2: Yes

2. Has the statistical analysis been performed appropriately and rigorously?

Reviewer #1: N/A

Reviewer #2: N/A

3. Have the authors made all data underlying the findings in their manuscript fully available (please refer to the Data Availability Statement at the start of the manuscript PDF file)?

Reviewer #1: Yes

Reviewer #2: No

4. Is the manuscript presented in an intelligible fashion and written in standard English?

Reviewer #1: Yes

Reviewer #2: Yes

5. Review Comments to the Author

Reviewer #1: This is a good qualitative study on communication, inclusion, and psychological wellbeing among deaf and hard of hearing children. There is a good discussion on what sort of factors contribute to the psychological wellbeing and inclusion of Deaf and hard of hearing (DHH) children. The topic is interesting, and it seems the authors have spent some good time conceptualizing it. In addition, the general overview, and application of the study and its relevance in social contexts is missing.

The following can help improve the manuscript.

Abstract

The last bit of the abstract can be improved to better state the recommendations and implications of the findings.

Introduction

There is room for improvement in this section.

1.Authors first must state the broad theme or topic of the study. After introducing the broad theme/topic, its academic and practical significance (if applicable) must be discussed. You should present a convincing response to the question, "Why should anyone care about this article?" Author has already presented some data regarding this, but it is not enough evidence for justification of the study.

2. The author should indicates some gaps, inconsistencies in the literature that the current study has addressed and should explain the study’s main contribution .

Materials and Methods

The method used is appropriate for the type of study presented in the article.

1. How the sample size (35) has been determined is not described, author should provide some scholarly reference here.

2.It is mentioned that the interviews with the school students, caregiver and teacher were In-depth Interviews, and three Focus Group Discussions were held with DHH adults and disability leaders, mental health specialists.

The author should explain explained why different interview techniques were used for different interviews.

3.Much more could have been written about the validity and reliability of the study.

 

Results and Discussion

The results section gives an interesting description of a number of themes that have most likely had an important impact on the studies of the interviewed students.This section is well written.

1."Communication, inclusion and psychological wellbeing among deaf and hard of hearing children: A qualitative study in the Gaza Strip" According to the title of your study, psychological wellbeing among deaf and hard of hearing children is a key factor of this study. But I did not find enough writing or analysis about ‘psychological wellbeing’ in the result section.

Conclusion

I suggest the authors club the conclusion, limitations of the study, and future prospectus and elaborate it under "Conclusions or concluding remarks" as a heading.

Reviewer #2: Review of "Communication, inclusion and psychological wellbeing among deaf and hard of hearing children: A qualitative study in the Gaza strip"

This is a very important study, giving voice to deaf and hard of hearing (DHH) children, adults and their advocates in the Gaza strip. By analyzing a sufficient sample of interviews and focus groups, and nicely presenting the findings, the authors point at the difficult situation of this population in terms of language deprivation, stigma, and lack of basic services. They indicate needs and suggest solutions, making this study valuable for policy makers and anyone aiming at improving the situation for this population. I would also like to commend the authors for including DHH people from Gaza strip in the steering committee of the research, thereby strengthening its rigor.

I have several comments and suggestions to improve the manuscript.

1. The introduction is well-written. Under 'Materials and Methods' (p. 4 line 84), please summarize the exact number of participants involved in the study at each category (interviews with children, caregivers, teachers, focus groups, and how many participants per each focus group).

2. Sentence 104 starting with 'Referral systems' needs further clarification.

3. Appendix S1: Please provide the following information about the DHH children: Usage of hearing aids or cochlear implants, the age of fitting/implantation (if known), and communication mode (oral, sign, bilingual).

4. Data collection: Were the interviewers structured or half-structured? Please provide the full topic guides of the interviews and the focus groups, in the text or as an appendix.

5. Data analyses: This section is relatively weak. Please provide a detailed explanation of the different stages of TA that you conducted according to Braun and Clarke, who, and how many people were involved at each stage. If more than one person was involved in determining codes/categorizations, at what stage were these analyses compared and discussed, and how were discussions solved?

Results and Discussion

1. The authors mentioned in the introduction the unique situation of children in conflict zones, such as suffering from multiple stressors and traumas. They also mentioned that the research was carried out during the COVID-19 pandemic. Yet, these topics were not discussed in the study itself. Were there any data stemming from the interviews which could inform us on the experience of coping with hearing loss in these contexts? Were participants asked about conflict/pandemic issues? If no data exists, a reflective notation may help the readers understand why certain topics were not discussed with participants or not raised by them.

2. My overall impression of the study is that it advocates for the use of sign language in DHH children, their families, teachers, and community. While this is a very important recommendation, another direction exists as well, which is improving the oral-communication services and devices provided to DHH children and adults. These may include intensive speech therapies for children using hearing aids or cochlear implants (without which they can hardly benefit from them), auditory specialists (e.g., who can fit CI's), systematic newborn hearing screening, providing and informing on accessibility devices such as FM systems and lived transcriptions. These can be emphasized in more details both at the 'assistive technology and communication' section (e.g., line 309 onwards), and at the discussion and recommendations.

3. Maybe related to that, the authors mention in the discussion the relatively positive situation of children in special schools, who use sign language, are engaged with DHH peers and seem to be at lower risk for poor mental health. This is of course important to note, yet, it is also important to consider in the discussion the word-wide dispute on schooling segregation and the role that special schools may at times play in segregating children from society and limiting their opportunities. It seems that in the current context at Gaza strip, special schools provide opportunities that lack in mainstream schools (which is well described by the authors), also because they are private. Yet, it is important to bring about a wider perspective on schooling options, based on experiences in other countries. Thus, adapting recommendations that strive at ideally providing DHH with all options, including full integration within mainstream schools (when done well).

4. Results line 353: Please mention the (apparently) lack of systematic governmental or organizational training and resources for ambulant teachers, who can instruct mainstream teachers about the needs of DHH children and provide DHH children with personal educational and other assistances (thereby supporting their inclusion in mainstream schools).

5. Line 405: When surveying detriment factors, emphasis needs to be given also to structural and material barriers (such as lack of governmental services), in addition to cultural and attitudinal ones.

6. Discussion line 598: Mental health services for DHH people would benefit also from recruiting signing mental health professionals, DHH professionals, and sign language interpreters specialized in mental health.

7. Finally, in your suggestions please provide an informative review of existing interventions designed especially for DHH children (e.g., at schools, families). These can include also interventions applied in high-income countries.

Editing suggestions:

1. Page 1 line 47: Please change to 'moderate to severe or profound (depending on your source)', instead of 'complete' hearing impairment.

2. Line 58: The word 'are' is redundant.

3. Line 88: 'and caregivers' is redundant.

4. Line 124: The word 'to' seems incorrect here.

6. PLOS authors have the option to publish the peer review history of their article (what does this mean?). If published, this will include your full peer review and any attached files.

**Do you want your identity to be public for this peer review?** For information about this choice, including consent withdrawal, please see our Privacy Policy.

Reviewer #1: **Yes: **Morshed Alam

Reviewer #2: No

---

## [Decision Letter · Decision Letter 1]

21 Apr 2023

Communication, inclusion and psychological wellbeing among deaf and hard of hearing children: A qualitative study in the Gaza Strip

PGPH-D-22-01952R1

Dear Mr Scherer,

We are pleased to inform you that your manuscript 'Communication, inclusion and psychological wellbeing among deaf and hard of hearing children: A qualitative study in the Gaza Strip' has been provisionally accepted for publication in PLOS Global Public Health.

Best regards,

Julia Robinson

Executive Editor

Reviewer Comments (if any, and for reference):

Reviewer's Responses to Questions

**Comments to the Author**

1. If the authors have adequately addressed your comments raised in a previous round of review and you feel that this manuscript is now acceptable for publication, you may indicate that here to bypass the “Comments to the Author” section, enter your conflict of interest statement in the “Confidential to Editor” section, and submit your "Accept" recommendation.

Reviewer #1: All comments have been addressed

Reviewer #2: All comments have been addressed

2. Does this manuscript meet PLOS Global Public Health’s publication criteria? Is the manuscript technically sound, and do the data support the conclusions? The manuscript must describe methodologically and ethically rigorous research with conclusions that are appropriately drawn based on the data presented.

Reviewer #1: Yes

Reviewer #2: Yes

3. Has the statistical analysis been performed appropriately and rigorously?

Reviewer #1: N/A

Reviewer #2: N/A

4. Have the authors made all data underlying the findings in their manuscript fully available (please refer to the Data Availability Statement at the start of the manuscript PDF file)?

Reviewer #1: Yes

Reviewer #2: No

5. Is the manuscript presented in an intelligible fashion and written in standard English?

Reviewer #1: Yes

Reviewer #2: Yes

6. Review Comments to the Author

Reviewer #1: Manuscript should be accepted

Reviewer #2: The authors have sufficiently revised the manuscript according to previous review.

7. PLOS authors have the option to publish the peer review history of their article (what does this mean?). If published, this will include your full peer review and any attached files.

**Do you want your identity to be public for this peer review?** For information about this choice, including consent withdrawal, please see our Privacy Policy.

Reviewer #1: **Yes: **Morshed Alam

Reviewer #2: No
